# Systematic Review of Smoking Cessation Interventions for Smokers Diagnosed with Cancer

**DOI:** 10.3390/ijerph192417010

**Published:** 2022-12-18

**Authors:** Kate Frazer, Nancy Bhardwaj, Patricia Fox, Diarmuid Stokes, Vikram Niranjan, Seamus Quinn, Cecily C. Kelleher, Patricia Fitzpatrick

**Affiliations:** School of Nursing Midwifery and Health Systems, Health Sciences Centre, Belfield, University College Dublin, D04 V1W8 Dublin, Ireland; nancy.bhardwaj@ucd.ie (N.B.); patricia.fox@ucd.ie (P.F.); diarmuid.stokes@ucd.ie (D.S.); drvikramn@gmail.com (V.N.); 2020beauparc@gmail.com (S.Q.); cecily.kelleher@ucd.ie (C.C.K.); patricia.fitzpatrick@ucd.ie (P.F.)

**Keywords:** smoking cessation, cancer, tobacco control, co-design, systematic review

## Abstract

The detrimental impact of smoking on health and wellbeing are irrefutable. Additionally, smoking is associated with the development of cancer, a reduction treatment outcomes and poorer health outcomes. Nevertheless, a significant number of people continue to smoke following a cancer diagnosis. Little is understood of the smoking cessation services provided to smokers with cancer or their engagement with them. This systematic review aimed to identify existing smoking cessation interventions for this cohort diagnosed with breast, head and neck, lung and cervical cancers (linked to risk). Systematic searches of Pubmed, Embase, Psych Info and CINAHL from 1 January 2015 to 15 December 2020 were conducted. Included studies examined the characteristics of smoking cessation interventions and impact on referrals and quit attempts. The impact on healthcare professionals was included if reported. Included studies were restricted to adults with a cancer diagnosis and published in English. No restriction was placed on study designs, and narrative data synthesis was conducted due to heterogeneity. A review protocol was registered on PROSPERO CRD 42020214204, and reporting adheres to PRISMA reporting guidelines. Data were screened, extracted in duplicate and an assessment of the quality of evidence undertaken using Mixed Methods Assessment Tool. 23 studies met the inclusion criteria, representing USA, Canada, England, Lebanon, Australia and including randomized controlled trials (9), observational studies (10), quality improvement (3), and one qualitative study. Hospital and cancer clinics [including a dental clinic] were the settings for all studies. 43% (10/23) of studies reported interventions for smokers diagnosed with head and neck cancer, 13% (3/23) for smokers diagnosed with lung cancer, one study provides evidence for breast cancer, and the remaining nine studies (39%) report on multiple cancers including the ones specified in this review. Methodological quality was variable. There were limited data to identify one optimal intervention for this cohort. Key elements included the timing and frequency of quit conversations, use of electronic records, pharmacotherapy including extended use of varenicline, increased counselling sessions and a service embedded in oncology departments. More studies are required to ensure tailored smoking cessation pathways are co-developed for smokers with a diagnosis of cancer to support this population.

## 1. Introduction

Leaving no one behind is the focus of the Sustainable Development Goals for improving the planet and the health and well-being of the population [1]. Tobacco control measures are a fundamental component in attaining SDGs. At the same time, global prevalence rates of smoking are decreasing, supported by effective tobacco control mechanisms established by the World Health Organization (WHO) Framework Convention on Tobacco Control (FCTC) [2] and subsequent MPOWER measures [3]. The FCTC uphold human rights [4] and children’s rights [5], yet despite the downward trend, 8.7 million deaths annually are attributable to tobacco; 1.2 million of the deaths result in nonsmokers, including 65,000 children, due to second-hand smoke exposure [6,7]. The mortality is a stark reminder of the annual death toll, as is the fact that only two countries (Brazil and Turkey) have implemented all high-level policy requirements of the 182 that ratified the FCTC [8]. 

In 2022 recent estimates indicate a rise in smoking rates among teenagers aged 13 to 15 years in 63 of the 135 countries [8]. Furthermore, global data on smokers (aged 15 years and older) identifies 942 million men, the majority living in medium or lower-income countries (LMICs), and 175 million women, 50% of daily smokers living in very high-income countries [9]. Global consumption of tobacco products is 20.3 billion (19.5–21.2) daily [8] (p. 2347); the highest smoking prevalence rates globally exist in China, India, Indonesia, the USA, Russia, Bangladesh, Japan, Turkey, Vietnam, and the Philippines [8] (p. 2341) [9].

Societal costs of smoking and tobacco-related harms go beyond mortality and morbidity statistics, acknowledged by the targets identified by the Sustainable Developments Goals [1]. The financial cost of smoking is estimated at $2 trillion, with 30% of the cost arising from healthcare and treating smoking-attributable diseases [6].

In their estimates of deaths attributable to smoking, 1.68 million [95% CI 1.56–1.81] represented deaths due to ischaemic heart disease, chronic obstructive pulmonary disease 1.59 million [95% CI 1.41–1.76], and tracheal, bronchus, and lung cancer (1.31 million [95% CI 1.20–1.43] [6] (p. 2438).

The NIH reports that smoking is the leading cause of cancer and cancer-related deaths, accounting for nearly 10 million deaths in 2020 [7,10]. This fact remains despite the global investment in cancer through the Moonshot initiative, the programme tasked with accelerating scientific discoveries in cancer, including data sharing and fostering greater collaboration [11,12]. Evidence suggests smoking cessation, even after a cancer diagnosis, can significantly reduce all-cause mortality and is associated with improved treatment outcomes [13,14,15]. The Moonshot initiative sought the adoption of a health systems lens to improve cancer-specific outcomes.

The challenge to reduce tobacco consumption in high-risk populations continues. Keyworth et al. [16] provides consistent evidence suggesting that more training in smoking cessation and supports are required to assist healthcare professionals’ engagement and better understand their perceived challenges. Conlon et al. [17] presents barriers for health care professionals engaging with smokers diagnosed with cancer, including a lack of knowledge and perceptions of the benefit to patients. Little is known about the evidence base for specific quit services offered to smokers following a cancer diagnosis, despite the positive impact of quitting and the availability of local and national smoking cessation services. This review of the international evidence base was undertaken as part of a funded study to develop a smoking cessation pathway tailored for people with lung, cervical, head and neck and breast cancers.

### Review Question

What smoking cessation interventions exist for smokers diagnosed with head and neck, breast, cervical or lung cancer?

## 2. Materials and Methods

A review protocol was registered on PROSPERO CRD42020214204 [18] and reporting follows PRISMA guidelines [19] (Appendix A).

The primary outcome was to assess the evidence base and describe the interventions targeted at smokers diagnosed with head and neck, lung, cervical or breast cancer. Reported outcomes include a description of interventions, methods of reporting referrals, quit attempts and impact on the number of cigarettes smoked, and reporting biochemical verification or self-reporting quit attempts. Secondary outcomes are reported if healthcare professionals’ data were included, and we report their attitudes, knowledge, facilitators or challenges to service provision.

### 2.1. Search Strategy

Search strategies comprised search terms both for keywords and controlled-vocabulary search terms MESH and EMTREE in four databases: PsychInfo, EMBASE (via OVID), PubMed (via OVID), Cumulative Index to Nursing and Allied Health Literature (CINAHL). The search was limited to 1 January 2015 to 15 December 2020 and restricted to English language. Reference lists of included evidence were checked for further relevant articles.

### 2.2. Eligibility Criteria

Studies reporting a smoking cessation intervention to support smokers diagnosed with lung, head, neck, cervical or breast cancers. All study designs (experimental, observational, and qualitative) were included. A rationale for widening our inclusion criteria for study designs was the lack of evidence published by Zeng et al. in their Cochrane review [20].

### 2.3. Selection of Studies and Data Extraction

Two authors developed the searches and search strings (DS & KF); all database searches were completed by one author (DS) (Search strategy Appendix A). Following de-duplication, references were uploaded into Covidence management platform [21] (KF), and three authors independently screened all titles and abstracts (KF, PF, NB). Full texts of all potentially eligible studies were independently reviewed by three authors (KF, PF, NB). Disagreements were resolved by discussion with a senior author (PFi). A data extraction form was developed and modified from previous documents used by authors (KF, CCK). Extracted data included study characteristics (title, lead author, year of publication, country, study setting, study design), description of the intervention, number and characteristics of participants, outcomes, quit attempt, duration of follow-up, sources of funding). Data from included studies were independently extracted by two authors (KF & VN). 50% of papers were extracted in duplicate. The remaining data extracted were checked and verified independently by a third author (NB). The study design (required for a quality review) was independently assessed and confirmed by two authors (KF & NB).

### 2.4. Assessment of Quality

Two review authors (KF & NB) independently assessed the quality of included studies using the Mixed Methods Assessment Tool MMAT [22]. No disagreements emerged, and no studies were excluded based on quality assessment (Appendix A). The MMAT is used widely and considered a valid indicator of methodological quality using instruments for either randomized or non-randomized, descriptive, or qualitative studies.

### 2.5. Data Synthesis

Meta-analysis was not possible due to heterogeneity in study designs, participants, outcomes, and nature of the interventions and no attempt was made to transform statistical data. The SWiM criteria [23] guide a narrative summary, with data presented in tabular format and subgroup reporting.

## 3. Results

We identified 6404 articles in electronic searches and two further articles via hand searching. After duplicates were removed, titles and abstracts of 4293 records were screen screened 4126 records were removed as they did not meet the inclusion criteria. In total, 167 full-text articles were selected for review. After an evaluation against our inclusion criteria, 23 studies are included in this systematic review (Figure 1). Multiple papers are reported for two studies, and data from Abdelmutti [24] are reported in Guiliani [25]. Concurrent and additional reporting for outcomes in Crawford [26] are in four papers [27,28,29,30].

### 3.1. Study Characteristics

Geographically we report 23 studies from five countries (Table 1), the majority from the USA [26,31,32,33,34,35,36,37,38,39,40,41,42,43,44,45,46], and Canada [24,47,48]. We report evidence from England [49], Lebanon [50] and Australia [51], and Nine studies report evidence from experimental randomised controlled trials [26,31,35,37,38,42,45,46,50]. Evidence is included from observational studies, namely cohort [24,33,34,43,48] and cross-sectional studies [44,47]; from mixed methods studies [32,40,51], quality improvement studies [36,39,41], and a qualitative study [49].

To provide a comprehensive review of the evidence, we report studies that include data for the specified cancers [head and neck, lung, breast, and cervical cancer] in addition to reporting additional cancers noted in studies. Ten studies report interventions for smokers diagnosed with head and neck cancers [34,35,36,37,39,45,47,49,50,51], three report interventions for thoracic/lung cancers [40,43,45] and one study provides evidence for breast cancer [41]. The remaining studies report interventions for multiple cancers, including those specified in this review [24,26,31,32,33,38,42,44,46].

### 3.2. Characteristics of Interventions

Studies included participants ranging from 11 to 2652 (Table 2a–c). The interventions described in the evidence vary in content and duration of follow-up (1 month up to 24 months). Most patients are recruited to studies and programmes in hospitals as inpatients or when attending cancer centres or clinics; Conlon [47] enrolled patients attending a dental clinic for oncology patients located within a specialist cancer centre.

Interventions ranged from low tech, including the provision of a survey questionnaire, a book, attending clinic or provision of leaflets and resources [32,35,37,39,40,46,49,50,51] or interventions including establishing and developing referral systems, pharmacotherapy including varenicline for extended periods, development and use of digital apps, increased counselling sessions including face to face, telephone, additional visits to counsellors, adapting and link to national smoking cessation resources including quit lines, website information and published resources [24,26,31,33,34,36,38,41,42,43,44,45,47,48]. Bricker et al. [31] and Rettig et al. [45] incorporated national mobile phone apps within their hospital-based smoking cessation programmes.

Recruitment and enrollment of patients included several options, attending for a first consultation, commencing treatment while awaiting surgery or participants who had a cancer diagnosis for up to 5 years and may have completed a treatment cycle. Seven studies identified recruitment for newly diagnosed patients [24,41,42,46,47,48,51] and three studies recruited patients to the smoking cessation programme at the pre-surgery phase following diagnosis [40,43,50].

Several developed smoking cessation interventions linked to national smoking cessation programmes and provided participants with information for websites, quit lines, text messaging services and use of quit smart apps [24,31,34,38,45,48,51]. Ten studies reported the availability of free or low-cost varenicline and pharmacotherapies to support quitting [24,26,33,34,36,42,43,45,47,48]. Two studies [33,36] report the availability of nicotine replacement therapy [NRT] for family members. The duration of pharmacotherapies provided free varied per study with Crawford et al. [26] providing up to 12 weeks and further extension of this up to 24 weeks [30]. Other incentives described were monetary and provision of an ipad [37,38,45] (Table 2a–c).

Development of electronic referral systems as part of a smoking cessation programme was reported [24,25,43] specifically identifying ‘opt out’ systems in seven studies, whereby the smoking status was routinely sought and recorded for all patients attending clinics [24,36,39,41,44,45,47]. Nolan [41] noted including an ‘asking and recording smoking cessation status at all touch points’ by all healthcare professionals interacting with patients as a key development for their programme. All who were identified as smokers received advice on quitting and were provided with an opportunity for referral to a smoking cessation programme. Where this system was described, it included the development of an electronic health record or embedding of smoking cessation services [24,36,42,44], and the system provided data on referrals, outcomes and follow up. A minority of studies recorded patients’ smoking status at each clinic/hospital visit—and those that did, the data were captured from the development of the intervention programme as part of a research study or following commencement of a quality improvement initiative [24,36,39,41,43,44].

A minority of studies report smoking cessation services embedded within oncology services providing a bespoke service for patients attending clinics. The practitioners were oncology healthcare professionals (HCP) who completed additional smoking cessation training. McDonnell et al. [40] described support for patients via oncology nurses trained in smoking cessation, support for patients and family members from therapists within oncology services [36,40].

### 3.3. Outcomes of Interventions-Quit Attempts

The outcomes of interventions varied with limitations associated with small sample sizes. Table 3a,b report outcomes, referrals and quit rates. Increased referrals linked to electronic referral systems were reported in four studies [24,25,36,41]. Several studies reported the impact on quit rates [24,27,32,33,38,42,43,45] with Charlot et al. [32] identifying a quit rate at 8 weeks of 50.1% and 44.3% at 3 months. Conlon et al. [47] reported quit rates of 14.8% at 3 to 6 months and 12 months. Almost 50% of current smokers reporting using no smoking cessation supports to help their quit attempt, another 23.8% reported using nicotine replacement therapy [47].

Rettig et al. [45] noted higher quit rates in the intervention group OR 4.83, 95% CI 1.31–17.76, and for those who were married. Lower odds of quitting were reported in patients with a history of depression, having a co-addiction, and experiencing mucositis during treatment. Higher pain scores and mucositis were associated with increased smoking rates. Rettig suggests a value in integrating smoking supports into cancer treatment.

Cinciripini et al. [33] reported improved abstinence in the intervention group compared to usual care at 9 months RR 1.31 95% CI 1.11–1.56, *p* = 0.001, while rates of quitting did not differ between patients with cancer and those who did not have cancer, the results present a high rate of abstinence. Crawford et al. [26] used varenicline pharmacotherapy to support quitting but did not report a difference in quit rates at 12 weeks. Subsequently, Schnoll et al. [30] reported point prevalence smoking rates that were not significantly different at 24 and 52 weeks (Table 3a). Price et al. [29] notes that there was no increase in depressed mood scores, and abstinence was associated with improved cognitive function. Overall evidence suggests [26,27,29,30] consistent evidence of the use of varenicline therapy for 24 weeks.

Simmons et al. [46] recruited patients who had relapsed following a previous quit attempt and evidence from the study identifies the positive impact on quit attempts for married or having a partner OR 2.23 (95% CI 1.01 to 4.90). In relation to increased counselling supports, Park [42] identified that increased counselling sessions n = 8 IQR 4 to 11 was associated with use of quit pharmacotherapies observed in treatment group V usual care (77.0% V 59.1%), OR 2.31 95% CI, 1.32–4.04; *p* = 0.003 (Table 3a,b).

### 3.4. Challenges Experienced during Quit Attempts

Ghosh et al. [37] suggests that smoking was valued more by participants than any incentives to quit, noting the particular challenges for those with head and neck cancer and their difficulty travelling a distance to attend a specific smoking cessation clinic. Difficulties impacting successful quit attempts included winter weather, scheduling appointments for late time of the day and the effect of traffic or no parking at hospitals when attending appointments. While electronic or opt out referrals systems generated increased referrals, Guiliani et al. [25] reports low interest in referrals to subsequent smoking cessation programmes. They suggest this may be due to low motivation, lack of confidence to quit, stigma with smoking and having cancer, and fear that care could be impacted if refused referral. Other barriers identified services only available in English and a lack of longer term follow up.

Abdelrahim et al. [49] supports Guiliani et al. [25] as they identified challenges experienced by smokers with cancer trying to quit. Four themes emerged including guilt, barriers to quitting, the teachable moment, and identifying social motivation to quit and smoke. Evidence from this study suggestions that relapse is common, patients who refuse to quit did not indicate refusing overall—they wanted to be asked again. The obstacle was dealing with life and lack of control with cancer diagnosis, they wanted to control if and when they would quit smoking. Sustained support and encouragement are necessary as feelings of guilt and self-blame were reported. Barriers to quitting included the cost of NRT and pharmacotherapies in addition to cancer treatments. Participants with a diagnosis of head and neck cancer identified that side effects of treatment were relieved by smoking—provided relief. They suggest a teachable moment exists at diagnosis and this is supported by Conlon et al [47].

Social support was complex in both being a factor to encourage quitting, but reduced support could result in isolation and increased smoking [46]. Higher smoking rates were reported among those without social support and higher episodes of fatigue, depression, pain, and lower confidence to quit. Abdelrahim et al. [49] noted social support could increase smoking if a partner or friend is a smoker (Table 3a).

### 3.5. Healthcare Professionals Outcomes

Table 3b highlights outcomes for healthcare professionals from two studies [39,41] in this review. Ma et al. highlights discrepancies in practice with a lack of knowledge, lack of recording of the smoking status of patients, lack of awareness of smoking cessation services, competing life events, and lack of indoor smoke restrictions in own homes. The sample in the study is small. Despite this limitation, an improvement in staff knowledge is noted. Consistent challenges are reported by Nolan et al. [41], indicating personal barriers and assumptions made by practitioners in limiting their engagement with smoking cessation conversations for patients with cancer; and their perception of limited use of the pre-surgical period for commencing smoking cessation discussions.

Table 4 presents a summary of evidence indicating critical components and processes for each of the 23 studies in this review, combining the complex data from Table 2a–c and Table 3a,b We identified 13 studies providing successful outcomes, five of the studies reported evidence of increased referral rates but had limited impact on quit rates. Comprehensive development of interventions is critical given the complexity of care required.

## 4. Discussion

The evidence in this review foregrounds the range of processes and services offered by smokers with cancer (Table 4). The variation of methods and success of sustained pharmacotherapies and counselling [26,42] and limited success of many other interventions supporting quitting. What is known is that approximately 69% of smokers express a desire to quit [52]; however, limited engagement with smoking cessation services exists for smokers diagnosed with cancer. Despite what is known, few interventions are routinely implemented [53].

This review presents inconsistent outcomes from a comprehensive array of study designs [including pilot and feasibility studies], interventions and quality improvement programmes developed for smokers diagnosed with cancer. Interventions adopted in cancer centres range from the provision of information sheets—to the development of smartphone applications and the link to national smoking cessation programmes. This review comprehensively reports the characteristics of interventions and their success or other positive change.

Several systems issues are identified as critical in supporting smoking cessation programmes, including developing and using electronic patient records and recording patients’ smoking status at each clinical encounter (Table 2a–c and Table 4). The evidence in this review highlights the variety of approaches and timing of conversations from referral to pre-surgery to during therapy and post-therapy [41,42,46,47]. Abdelrahim et al. noted the importance of the duration of smoking cessation services and conversations provided and a need for information about the effect of continued smoking on recovery from cancer [25,40]. Conlon et al [47] suggests information about quitting should be consistently offered at all stages of the cancer journey. Identifying the timeline for introducing conversations on quitting was highlighted, with the importance of quitting smoking as a pre-surgical intervention. Table 4 presents evidence that suggests integration of smoking cessation services in cancer care as usual practice, developing electronic referral systems, increased consultation and follow-up reviews, and pharmacotherapies [free or low cost] are essential components of a smoking cessation intervention.

Cinciripini et al. [33] supports inventions in the oncology setting to enable sustained abstinence for patients with cancer and survivors; Rettig et al. [45] suggests the integration of services in oncology is part of the National Cancer Institute Cancer Moonshot initiative [11]. Croyle et al. [54] considers this a high-risk population group, and programmes for cessation should be developed. 

In this review, evidence identified significant barriers for healthcare professionals supporting smokers with cancer due to a lack of confidence and perceptions [39]. The findings in this review are consistent with Feuer’s [55] review of 29 studies that examined smoking relapse for a population who are cancer survivors. They suggest that smoking cessation after a cancer diagnosis is understudied, and that interpretation of interventions is challenging due to heterogeneity (p. 102237).

Research is required to determine optimal smoking cessation support after treatment ends, the duration of interventions, and cancer-specific tailoring [53,56]. Acknowledging the challenges of patients adjusting to chemotherapy regimens and diagnosis suggests a need to support sustained quit attempts and support patients who relapse to enable confidence-building and reduce stigma. It is crucial to consider the longevity of support that may be needed beyond the usual eight or 12-week programme. Santi et al. [55] confirm the challenges for those quitting and the potential for additional support beyond six months, as noted in the updated National Institute for Health and Care Excellence (NICE) guidelines [57]. Evidence indicates the importance of married/partnered patients for social support when quitting [53]. It is vital to ensure that smoking cessation programmes are inclusive and developed for diverse populations. Consistent evidence on developing population quit programmes is known [58]; what is evident is a lack of adaptability to smokers who have cancer and have different types of cancer, which may result in a variety of side effects and experiences from both cancer and cancer treatments.

The location of quit services is critical. There is evidence in this review supporting embedding smoking cessation services within oncology services (Table 4), and Ghosh et al. [37] presents the challenges for those who have to travel to attend clinics, including time and financial costs. Alternative methods reported in this review include engaging with national smoking cessation services and providing choices to patients on the mode of delivery, including online, email or by phone. The impact of COVID-19 provided opportunities for developing virtual clinics, and this may be a useful addition.

Warren et al. [59], almost ten years ago, identified the use of an automated referral system for those with a diagnosis of cancer into smoking cessation services. Conlon et al.’s [17] systematic review of healthcare professionals also confirms the importance of electronic referral systems, the responsibility of all health care professionals working in oncology to document smoking status, and the need for dedicated referral to a specialist smoking cessation advisor. Evidence in this review confirms the benefit of electronic referral systems and integrated health records.

### 4.1. Quality Review

Quality was assessed in twenty studies, as three studies were quality improvement programmes. Factors associated with lower quality of evidence include the reliance on self-reporting of smoking history, recall bias, and convenience sampling (Appendix A). However, confirmation of quitting was reported in several studies via biochemical verification and carbon monoxide monitoring [26,33,38,42,43,45,46,51]. Several studies included randomisation and a follow-up period of 24 months with statistical adjustment for independent factors, including gender, age, marital status, and comorbidities (Table 3a,b). We did not remove any study from this review, and we acknowledge the breadth of evidence includes studies with a lower quality design, with short-term follow-up of interventions; however, we wished to be comprehensive and acknowledge the tailored responses adopted to support smokers who have cancer from feasibility or pilot studies.

### 4.2. Limitations in the Review Process

A key strength of this review is that it addresses a knowledge gap and has collated evidence from a broad methodological base to report the interventions adopted to support smoking cessation in smokers with cancer. Due to the heterogeneity of included studies, meta-analysis was not performed, while the descriptive nature of studies prevents identifying a causative relationship between measures and outcomes. The duration of interventions varied in the evidence presented, with five studies reporting interventions of less than six months duration [32,38,39,50,51]. We did not exclude studies based on the duration of intervention to present the extent of evidence for this population group. We acknowledge that while a summary of characteristics and outcomes is presented, insufficient evidence is available to statistically evaluate and summarise the relationship between individual measures, and further studies are required to elucidate this. Despite this, the systematic approach to this review has identified the scope of interventions implemented to date for a critical population of smokers diagnosed with cancer. This adds to the limited body of evidence published [17,20,56].

We acknowledge reporting bias from self-reported data and the financial incentives reported in studies. Publication bias was minimised with follow-up contacts with authors for early reporting and by including observational study designs. In addition, we have presented overlapping papers and identified singular primary studies for completeness. We report studies published in English, searching four databases and a timeline of six years, as this was one component of an ongoing research study. However, the methods used to complete this review were complete and adhered to Cochrane review standards [60].

We acknowledge that patient and public involvement (PPI) was not featured in any study included in this systematic review. We do report that PPI was a component of this systematic review. Two experts by experience from the study steering committee were invited to review the preliminary evidence and assist in developing our understanding and reporting as co-authors. Secondly, a summary of the review evidence was presented as part of a PPI Patients Voice in Cancer workshop (April 2022) [a component of the broader research study].

## 5. Conclusions

This novel systematic review summarises the evidence base to date, identifying specific factors for consideration in the development of a smoking cessation intervention targeted at smokers who have cancer. As reported by participants, the nuances and sensitivities of the population, the impact of treatment, and the stigma associated provide a strong voice for their input in the subsequent co-development and implementation of interventions to support tobacco control and smoking cessation in cancer care services. A one size fits all approach to development is unhelpful, and further research should embed public and patient involvement at the outset.

## Figures and Tables

**Figure 1 ijerph-19-17010-f001:**
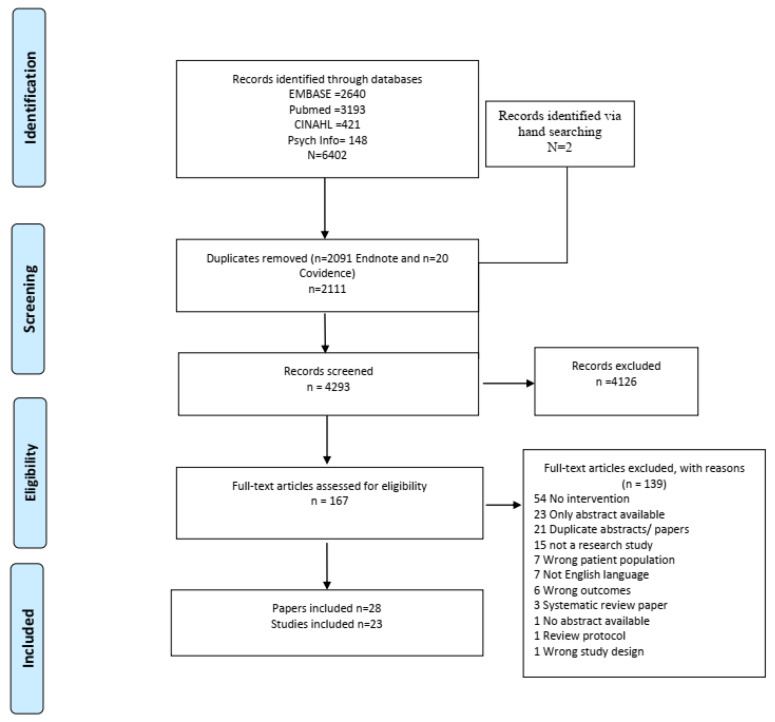
PRISMA Flowchart.

**Table 1 ijerph-19-17010-t001:** Study Characteristics.

Study ID	Country	Study Design	Setting	Population	Cancer Type	Linked Additional Papers
Abdelmutti et al., 2019 [24]	Canada	Cohort	Cancer Centre	Patients with cancer	Breast, CNS, Endocrine, Eye, GI, genitourinary, Gynae, head and neck, Haematology, lung, melanoma, sarcoma	
Abdelrahim et al., 2018 [49]	England	Qualitative	Hospital	Patients with cancer	Head and Neck	
Bricker et al., 2020 [31]	USA	Pilot double-blinded RCT	Cancer centres	Patients using a Phone app	Lung Breast Skin Cervical Colorectal Leukemia Non-Hodgkin lymphoma Pancreatic Esophageal Liver Prostate/Stomach/Throat/All others	
Carroll et al., 2019 [27]	USA	Secondary analysis of data from 12-week trial data NCT 01756885	Hospital	Patients with cancer	Genitourinary Breast Skin Lung Haematological Head and neck Kidney/liver/pancreatic.	Reported in Crawford 2019 [26]
Charlot et al., 2019 [32]	USA	Mixed methods	Medical Centre	Patients with cancer	Lung, breast, colon, prostate, leukaemia, melanoma/skin, other	
Cinciripini et al., 2019 [33]	USA	Prospective cohort	Hospital	Patients with cancer Employees Patients with no history of cancer	Breast Lung Head and neck Colorectal and other GI Prostate/Other genitourinary, Lymphoma and other hematologic, Melanoma and other skin/Other (cancers <2% of total).	
Conlon et al., 2020 [47]	Canada	Cross-sectional prospective cohort	Cancer Centre Dental Oncology clinic	Patients with cancer	Head and neck	
Crawford et al., 2019 [26]	USA	Two group ITS NCT 01756885	Hospital clinics	Patients with cancer	Head and Neck, Lung, haematological, breast, GI, genitourinary, Skin, kidney/pancreas/liver	Additional papers linked Carroll 2019 [27] Miele 2018 [28] Price 2017 [29] Schnoll 2019 [30]
Davidson et al., 2018 [48]	Canada	Prospective cohort	Hospital	Patients with cancer	Not specified	
Day et al., 2020 [34]	USA	Retrospective cohort study	Hospital	Patients with cancer	Head and neck	
Foshee et al., 2017 [35]	USA	Pilot RCT	Hospital	Patients with cancer	Head and neck	
Gali et al., 2020 [36]	USA	Quality Improvement study	Hospital clinics/cancer care centre	Patients attending cancer clinics and their family members	Head and neck Thoracic oncology GI surgery oncology	
Ghosh et al., 2016 [37]	USA	Prospective 2 group pilot RCT	Hospital clinics	Patients at risk of head and neck cancer and those previously diagnosed	Head and neck	
Giuliani et al. 2019 [25]	Canada	Interrupted time-series study	Cancer centre and largest single-site cancer hospital	Patients	Not specifically noted	Reported study Abdelmutti 2019 [24]
Krebs et al. 2019 [38]	USA	Pilot RCT	Hospital clinic	Patients with cancer	Gastrointestinal Lung Urologic Colorectal Gynecologic Other	
Ma et al. 2016 [39]	USA	Quality improvement project	Hospital	Patients and staff	Head and neck cancer	
McDonnell et al., 2016 [40]	USA	Prospective Mixed methods	Hospital clinic	8 family dyads. (Patient and a family member)	Thoracic patients with known or suspicious neoplasms	
Miele et al., 2018 [28]	USA	Cross-sectional secondary data analysis from RCT	Hospital clinics	Patients	Head and neck, lung and ‘ other sites’ reported	
Nolan et al., 2019 [41]	USA	Mixed methods Quality Improvement programme	Hospital clinic	Patients with cancer and health care providers	Breast cancer	
Park et al., 2020 [42]	USA	RCT	Hospital and cancer centre clinics	Patients	Thoracic Breast Genitourinary, Head and Neck, Gastrointestinal. Lymphoma Gynaecological Melanoma	
Phillips et al., 2020 [43]	USA	Retrospective review of Database—Cohort study	Hospital	Patients with cancer.	Lung cancer	
Price et al., 2017 [29]	USA	RCT 756885	Academic hospital centres	Patients with cancer	Genitourinary, Breast, Lung, Skin, Hematological, Head and Neck, Gastrointestinal, Kidney, Pancreas, and Liver	Reported in Crawford 2019 [26]
Ramsey et al., 2020 [44]	USA	Cross-sectional, retrospective	Outpatient clinics	Patients with cancer	Not specific	
Rettig et al., 2018 [45]	USA	Pilot RCT	Cancer centre	Patients with cancer	Head and neck. Thoracic	
Schnoll et al., 2019 [30]	USA	RCT 756885	Cancer centres	Patients with cancer	Genitourinary, Breast, Lung, Skin, Hematological, Head and Neck, Gastrointestinal, Kidney, Pancreas, and Liver	Reported in Crawford 2019 [26]
Simmons et al., 2020 [46]	USA	RCT 01630161	Hospital Cancer centre	Patients with cancer	Thoracic, Head and Neck, Gastrointestinal, Breast, Genitourinary, Gynecological, Hematological, Cutaneous, other	
Smaily et al., 2021 [50]	Lebanon	RCT	Hospital	Patients HNC	Head and neck cancer	
Smith et al., 2019 [51]	Australia	Mixed methods	Hospital	Patients HNC defined as smoked in last 30 days prior to diagnosis	Head and neck cancer	

**Table 2 ijerph-19-17010-t002:** (**a**) Description of interventions from Randomised Controlled Trials. (**b**) Description of interventions from observational studies. (**c**) Description of interventions from mixed methods and qualitative studies. (**d**) Description of interventions from quality improvement studies.

**(a)**				
**Study ID**	**Population**	**Describe/Type of Intervention/Description of Smoking Cessation** **Services**	**Comparison (If Applicable)**	**Duration of Intervention**
Bricker et al., 2020 [31]	Patients	Quit2Heal designed to help cancer patients stop smoking by providing skills to cope with cancer-related shame, stigma, depression, anxiety, and cancer-specific health consequences of continued smoking versus quitting. Uses a personalised quit plan–approved cessation medications they can obtain on their own, users are taken to the home screen where uses progress through all 9 levels of the intervention content, receive on-demand help in coping with smoking urges, track the number of cigarettes smoked daily, and track how many urges they let pass without smoking. The program is self-paced, and the content is unlocked in a sequential manner. For the first 5 levels, exercises are unlocked immediately after the prior exercise is complete. For the last 4 levels, the next level will not unlock until users record 7 consecutive smoke-free days. If a participant lapse (e.g., records having smoked a cigarette), the program encourages (but will not require) the participant to set a new quit date and return to the first 5 levels for preparation.	NCI’s Quit Guide app. Quit Guide is a non-targeted smoking cessation app designed with 4 sections, thinking about quitting, preparing to quit, quitting, staying quit. for the general population of smokers, with 4 sections of content.	12-month duration of intervention.
Crawford et al., 2019 [26] Follow up Carroll et al., 2019 [27] Schnoll et al., 2019 [30]	Patients	Patients identified by electronic databases at cancer centres. Patients screened, initially by telephone and in-person, for study interest and eligibility. The study used data from a clinical trial that randomised cancer patients to 12 weeks of varenicline (open-label) followed by either 12 weeks of placebo or 12 weeks of varenicline; all participants received smoking cessation counseling sessions (NCT01756885). Participants received a 60 min in-person counseling session when they began medication at Week 0 and then four counseling sessions at Weeks 1, 4, 8, and 12. Assessments were conducted and carbon monoxide breath samples were collected in-person at Weeks 0, 4, and 12. For the present analyses, only data up to 12 weeks were used.	12 weeks of varenicline (open-label) & 24 weeks counselling. (follow up either 12 weeks of placebo or 12 weeks of varenicline). Carroll 2019 [27] Standard care 12 weeks of open-label varenicline & 24 weeks of behavioural counselling.	12 weeks Carroll 2019 [27] 12 weeks treatment and 24 weeks counselling duration
Foshee et al., 2017 [35]	Patients	Patients were randomised into two treatment groups: Intervention group received a free copy of The Easy Way to Stop smoking. Both groups received smoking cessation counselling at recruitment. Follow up surveys at 2 weeks to six weeks and 6 mths to 1 yr via phone.	Second group advised to purchase book The Easy Way to Stop smoking. Received smoking cessation counselling at recruitment.	Follow up duration 6 months to 1 year
Ghosh et al., 2016 [37]	Patients (veterans)	Voluntary enrollment. Randomised into cash incentive $150 at follow up V no additional incentive. All enrolled to smoking cessation classes—3 classroom session. Payment of $50 for each class attended for both groups. Follow up 30 days history including smoking status, confirmed by biochemical verification CO breath test. If negative for tobacco $150 paid to intervention group. Patients contacted by phone 10 occasions before considering lost to follow up. All patients completed SF Health Survey. 3-month assessment—repeat assessments and biochemical verification by urinalysis. Payment $150 if tobacco negative for intervention group. 6 months follow up repeat of 3 months and $150 for intervention group. In total $600 paid to intervention group.	Three smoking cessation classes offered,, follow up 30 days, 3 months, and 6 months. Payment for attending cessation classes	6 months follow up
Krebs et al., 2019 [38]	Patients	Intervention: Standard Smoking cessation care (4 telephone or bedside counselling sessions, and in in house information material. Sessions from oncology nurses who were tobacco specialists included: (1) motivation building, choose quit date, review print information, provide information on cessation pharmacotherapies. (2) coping with smoking urge/prevent relapse, 3 and 4 sessions) focus on relapse prevention or recycling to quit attempt for those who resumed smoking. Checklist completed by providers to track adherence and fidelity. And QuitIT (Smoking cues coping skills game using social cognitive theory of 10 episodes and 9 situations/smoking triggers) an app installed on iPad. Patients trained to use games, rules, objectives, watch video tutorials and RA evaluated patients’ comprehension. Encouraged to play game 3 to 4 times weekly for 1 month. Patients also received coping cards—resembling playing cards with strategies from the game. Patients were loaned iPad for 1 month and Could contact RA if technical issues.	Standard care only as described.	1 month follow up
Park et al., 2020 [42]	Patients	Intervention Participants were offered 4 weekly telephone counseling sessions, 4 biweekly telephone sessions delivered over 2 months, and 3 telephone booster sessions monthly. The number/length of sessions based on pilot work with patients from the Massachusetts thoracic oncology clinic. Participants offered a choice of 12 weeks of FDA-approved smoking cessation medication at no cost; they were not required to use any medication. The medication selected by participants was prescribed in the EHR and dispensed (in person or mailed). Participants received an initial 4-week supply of cessation medication (varenicline, bupropion sustained release, single or combination nicotine replacement therapy/patch and/or lozenges) with the option of receiving up to 2 additional 4-week supplies. Based on Self-regulation model and Health Belief model.	Participants offered 4 weekly telephone counseling sessions plus education and cessation medication advice. 1st session, tobacco counselors used a decision aid, designed for patients with cancer, to make medication recommendations.	Follow up 3 months and 6 months.
Rettig et al., 2018 [45]	Patients	The intervention group had the same study visit schedule, surveys, exhaled carbon monoxide administration, and mental health screening as the control group. 8-week programme. There were up to 4 additional daily visits during the first week. At baseline, the intervention group received the smoking cessation workbook and underwent intensive tobacco treatment specialist motivational interviewing, with brief follow-up motivational interviewing sessions at subsequent study visits, daily for the first week, weekly for 8 weeks. Other additional interventions received included: enrollment in the National Cancer Institute’s free smokefreetxt text-messaging program (smokefree.gov); contingency management at each visit, by which participants received $5 gift cards for biochemically confirmed smoking abstinence; and guided pharmacotherapy. Pharmacotherapeutic options offered were combination nicotine replacement therapy (patch/gum, patch/lozenge, or patch/nasal spray), bupropion, and varenicline. Participants receiving combination nicotine replacement therapy were instructed to use the patch daily, and to also use the nicotine gum, lozenges, or nasal spray as needed. Medication recommendations were based on mental health screening, comorbidities, and allergies, & oversight by a physician with expertise in tobacco cessation. Over-the-counter nicotine replacement therapy was provided for free (estimated cost per participant $240), and prescriptions provided for other medications. Participants were permitted to opt-out of intervention components.	4 intervention components for usual care: (1) brief counseling by a trained tobacco treatment specialist consistent with the “5 As” (2) a smoking cessation workbook tailored to patients with cancer patients; (3) contact information for local and national smoking cessation resources, including some offering free nicotine replacement therapy; and (4) mental health screening to evaluate depressive symptoms. At every visit, surveys were administered to ascertain smoking behavior and use of smoking cessation resources, and exhaled carbon monoxide testing. All participants were offered small gift cards for each completed study visit ($10 at baseline and $5 at follow-up).	weeks 1 to 8, 3, 6, 12 months
Simmons et al., 2020 [46]	Patients	Intervention group Brief single session with Tobacco Treatment Specialist. Use 5As and provided with pharmacotherapy if required. Received DVD educational support, plus a validated self-help interventions for preventing relapse. Forever free intervention—8 booklets	Brief single session with Tobacco Treatment Specialist. Use 5As and provided with pharmacotherapy if required.	Follow up 2, 6, and 12 months
Smaily et al., 2021 [50]	Patients	Intervention group: 10–15 mis counselling on smoking cessation based on 5 As. 8 weeks Nicotine patch with dosage tailored to dependence (Fagerstrom score). Participants provided with information in books, websites on smoking cessation, side effects and offered a free mobile phone app for support and provided with a hotline number for emergencies. All received a follow up phone call on week 5 to record compliance.	Usual care group: Brief advice.	8 weeks
**(b)**				
**Study ID**	**Population** **Study Design**	**Describe/Type of Intervention/Description of Smoking Cessation** **Services**	**Comparison (If Applicable)**	**Duration of Intervention**
Abdelmutti et al., 2019 [24]	Patients (Cohort)	Developed CEASE screening and referral tool: clinically supported electronic tool—e referral system. Adopted from Ottawa Model of Research Use knowledge translation framework. Based on 3As (ask, advise, act). Developed to introduce SC screening, educate patients and families on quitting, increase awareness of HCPs. CEASE tool embedded in existing assessment program/use of electronic tables in clinics. Monthly audit reports completed and use of traffic light system comparing clinics. Patient and provider education required. Guiliani 2019 [37] CEASE smoking cessation programme. Smoking Cessation e referral system. The Cancer Care Ontario agency oversees quality of care and CAN–ADAPTT smoking cessation service. Intervention was point of care assessment and review electronically for each new patient attending and diagnosed with cancer. Patients were assessed using CEASE (delivered electronically on a tablet) which had 3 components: (i) Ask/Assess—patient-reported smoking status (ii) Advise—standardised education regarding smoking (iii) Assist—patient-directed automated referral to patient’s choice of smoking cessation service. They are provided with referral to smoking cessation or resources via use of Tablet. 5As incorporated with calls within a week to follow up with the patient directly. Data linked to patient record.	Introduced for all patients. Those who decline referral—documented inpatient record, ‘decline referral or chose to quit on their own.	2 years presented. Giuliani 2019 [25]—6 months pre-implementation, 8 months during transition, 6 months post-implementation (June to November 2016)
Cinciripini et al., 2019 [33]	Patients, Employees Patients without cancer (Cohort)	Tobacco Treatment Programme consisted of an initial in-person consultation (60–90 min), plus 6 to 8 subsequent follow-up treatment sessions (30–45 min) over an 8- to 12-week period. 95% conducted by phone. Treatment involved behavioral counseling for smoking cessation and other psychological or psychiatric intervention, based on principles of Motivational Interviewing. Patients received 10 to 12 weeks of pharmacotherapy including nicotine replacement (patch or lozenge), bupropion, and varenicline, either alone or in various combinations. Each treatment plan was personalised, e.g., counseling sessions, duration, content, and choice of pharmacotherapy.	None. All received intervention	9 month follow up for participants
Conlon et al., 2020 [47]	Patients (Cross sectional)	Individualised counselling for smoking cessation by staff trained and certified under Centre for Addiction and Mental Health Training Enhancement in Applied Counselling. Use 3 As. All smokers offered individual intensive clinical counselling with pharmacotherapy or NRT. Follow up consultations 1–2 weeks after intervention. Follow-up for Tobacco intervention weekly during active radiation therapy and post-treatment follow up 4, 8, weeks, 6 months and 1 year.	All smokers offered individual intensive clinical counselling with pharmacotherapy of NRT	Up to 1 year
Davidson et al., 2018 [48]	Patients (Cohort)	2014 Smoking Cessation Program for all cancer patients who use tobacco, based on the Ottawa Model for Smoking Cessation. All patients registering before first consultation complete a brief questionnaire about their smoking status, including the preceding 6 months. Data collected on smoking history and time to first smoke. Patients asked if wish to be referred to Smoking Cessation Programme. At referral are provided with a Quit Kit contains information from Cancer Care Ontario and the Ottawa Model for Smoking Cessation and the various options available to assist them with smoking cessation. Within 2 weeks, a smoking cessation champion (a nurse or radiation therapist, with additional training in smoking cessation) contacts the patient to provide additional information, counselling, and support, and to offer a referral to the Public Health Unit for further counselling and NRT. When patients are first screened and accept a referral to the Smoking Cessation Program, they are offered follow-up by automated telephone or email contact for 6 months. An automated telephone call is regularly made to the patient to assess the individual’s progress with smoking cessation. A nurse monitors the communication and contacts patients as needed for additional counselling or support.	None	6 months follow up
Day et al., 2020 [34]	Patients (Cohort)	MD Anderson Tobacco Treatment Programme (TTP) is a comprehensive and personalised intervention including counseling, pharmacotherapy, and management of mental health conditions. TTP participants begin with an in-person consultation followed by 6 to 8 treatment appointments completed within 8 to 12 weeks. Abstinence data prospectively collected at 3-, 6-, 9-, and 12-month intervals beginning at enrollment. Abstinence is defined as 7-day point prevalence of cigarette abstinence at 9-month follow-up.	None	9 month follow up
Phillips et al., 2020 [43]	Patients (Cohort)	Tobacco treatment specialist (TTS). Opt-in programme. Scheduled face to face counselling arranged (free) if agreement. TTS counselling was 1 hr face to face using motivational interviewing and discuss treatment options. Patients chose type of pharmacotherapy: nicotine replacement (patch, gum, lozenges), bupropion, varenicline, or none. They were also advised to set a quit date. Follow-up with the TTS occurred per patient request and included face-to-face appointments, telephone conversations, or both. Carbon monoxide (CO) levels taken at preoperative visits and on the day of surgery. If CO level >11 ppm surgery could be cancelled or postponed. All patients were given unsupervised home exercise program and asked to complete at least 30 min of moderate exercise daily.	None	Follow up day of surgery, 2 weeks, 6, 12, 24 months.
Ramsey et al., 2020 [44]	Patients (Cross sectional)	ELEVATE smoking cessation module and smoking treatment engagement. ELEVATE programme introduced. ELEVATE facilitates systematic implementation of the “5 A’s” tobacco cessation intervention framework (Ask, Advise, Assess, Assist, Arrange) using EHR functionality to ensure consistent tobacco use assessment and cessation treatment support. ELEVATE is a paradigm shift from a cessation specialist referral model of care to a low-burden point-of-care treatment model. Patient is treated at regular point of care. Not referred to separate service. ELEVATE uses a two-pronged implementation approach that focuses on optimising the EHR-enabled workflow and evaluating practice data for feedback. This approach is underpinned by an internally developed Epic module specifically designed to facilitate end-to-end delivery of the “5 A’s,” Assist allows best practice advice—oncologist can prescribe cessation medication. Counselling offered every 90 days+ cessation meds if needed.	Quality improvement for all	10 months
**(c)**				
**Study ID**	**Population** **& Study Design**	**Describe/Type of Intervention/Description of Smoking Cessation** **Services.**	**Comparison (If Applicable)**	**Duration of Intervention**
Abdelrahim et al., 2018 [49]	Patients (Qualitative)	Completion of a survey or participated in a pilot smoking cessation service. Those that declined to participate in the pilot study invited, to explore the ‘difficult to reach individuals and provide representative understanding of views and experiences of the different groups of patients encountered in a head and neck follow up clinic. A Specialist nurse invited participants who were current/former smokers, had completed treatment for HNC and were in the follow-up stage of treatment. The intention was to recruit successful quitters and unsuccessful participants from the pilot study to explore the difference in their experiences and reasons they believe made them successful or not.	NA	Interviews held 12 months–4 years post treatment (intervention was 12 months)
Charlot et al., 2019 [32]	Patients (Mixed methods)	Feasibility and acceptability of a mindfulness-based smoking cessation (MBSC) medical group visit for low-income and racially diverse smokers with cancer. 8 weeklies 2 h medical group visit cofacilitated by clinical and MBSC practitioner. The curriculum components: a low literacy adapted mindfulness training programme tailored for people with chronic back pain. And You can quit smoking programme adapted from hospital tobacco treatment centre. MBSC included yoga, walks, stretching, massage. Participants provided with audio disc, info on body scans. Provided with manual and CD player if needed. Tobacco treatment included information on smoking, effects on body and pharmacotherapy information.	None. All received intervention (2 cohorts enrolled).	3 months
McDonnell et al., 2016 [40]	Patients, family members (Mixed methods)	Guided by social cognition theory and conflict theory. Tobacco-Free Family (TFF) intervention based on clinical practice guidelines from US Dept of Health. Includes quality decision making tutorial, three decision balance sheets focusing on three smoking-related decisions, and an evidence-based smoking cessation program and support (described in earlier article 2014—Each dyad received smoking cessation program booklet from hospital. Short counseling sessions (four face-to-face sessions delivered in hospital or clinic setting and up to six optional booster communications delivered remotely via telephone and/or the Internet) by a study coordinator, an oncology-certified nurse with training as tobacco treatment specialist, who provided additional information about nicotine dependence, quit date preparation, withdrawal symptom and trigger management, smoking cessation medications, weight control, exercise, stress management, and relapse prevention. A meditation CD provided. Flyers posted in workrooms, and pocket reference guide given to each team member to increase confidence using 5As.	None	Follow up collections at discharge, 1 month post op. 6 months post up
Smith et al., 2019 [51]	Patients (Mixed methods)	No intervention reported—the commencement of treatment for cancer was ‘intervention’. Follow up study of patients during treatment to identify factors that influenced quit rates.	Former/Never smokers	Surveys pretreatment, during treatment, 1 month and 3 months post-completion
**(d)**				
**Study ID**	**Population**	**Describe/Type of Intervention/Description of Smoking Cessation** **Services.**	**Comparison (If Applicable)**	**Duration of Intervention**
Gali et al., 2020 [36]	Patients Families	Automated opt-out at referral. Stanford Tobacco Treatment service. Patients are screened and provided with information and advised follow up phone calls in one week from cessation service. Patients can opt-out of receiving call. Up to 3 phone calls made with those who agree to opt-in. Messages sent via Electronic Health record introducing treatment specialist, tobacco treatment options (including e-cigs) avail to patient and family for free or reduced price. Menu of services includes counselling, and or medications. Counselling is in person before or after clinic visit, individually or group, or virtually or over phone. Counselling provided by clinical psychology students supervised by psychologists. Counselling is three sessions with the option to continue. Translation services provided. Toll free number if via phone in 6 languages. Information on NCT smokefree.gov, web, chat and phone support service provided. Cessation meds provided by virtual pharmacy with same-day delivery. Patients and family members offered 2-week free trial. Varenicline or bupropion prescribed following consult with physician, prescriptions written and filled on same day/delivery. To communicate metrics a member of Tobacco Treatment Service attends clinic morning huddle and or monthly staff meeting with feedback engagement data.	None	Varied and tailored (6 months up to one year)
Ma et al., 2016 [39]		For providers—a two-page teaching sheet developed for patients on smoking cessation. Information on tobacco, reasons to quit and information on resources to help quitting. 10 min presentation on resources and helplines provided to health care staff. Patients to be ask smoking status at each visit. Tobacco cessation discussion note template added to electronic patient record for staff documentation. Patients received a 2-page sheet. 4 weeks post-implementation, a survey of staff on knowledge of resources, review of medical record for frequency of tobacco cessation discussion documentation.	Quality improvement project changing practice.	Post QIP May to Sept 2014—5 months.
Nolan et al., 2019 [41]	Patients and employees	Opt-out referral system developed. Used Consolidated Framework for Implementation Research (CFIR). Clinical assistant confirmed smoking status during rooming/documenting. Nurse/Dr also asked the patient about smoking and provided advice on risks pre-op and referral to Nicotine Dependency Clinic (NDC) which was in place prior to new referral processes. All conversations documented in electronic record. Surgeons asked to reinforce messages. NCD programme: 5 practitioners available daily and can see patient on day of referral. 45 min consultation, up to 3 telephones follow up calls, Treatment specialists provide behavioural counselling and medication management overseen by physician.	None as new process implemented in clinic. Pre-implementation data	12 months

**Table 3 ijerph-19-17010-t003:** (**a**) Patient and Family Outcomes. (**b**) Healthcare professionals outcomes.

**(a)**					
**Study ID**	**Outcome Groups**	**Participant Demographic (Age, Sex, Smoking History If Included. Number Cigs Smoked; Prev. Quit Attempt)**	**Biochemical Verification**	**Outcome Measure**	**Outcomes**
**Randomised Controlled Trials**					
Bricker et al., 2020 [31]	Patients	Quit2heal Intervention group n = 29 Age 42.9 yrs (SD 12.0). Male 21% (6/29) White 79% (23/29) Married 48% (14/29) Working 52% (15/29) High school ed. Or less 41% (12/29) LGBT 7% (N = 24) Fagerstrom test score 5.1 (SD 2.4) Smokes more than half pkt cigs per day 59% (17/29) Smoked for 10+ years 90% (26/29) Use ecigs in past month 21% (6/29) One quit attempt in past 12 months 52% (15/29) Number of attempts to quit in 12 months 1.6 (SD 2.3) QuitGuide Control group n = 30 Age 47.3 yrs (SD 13.5). Male 30% (9/30) White 77% (23/30) Married 40% (12/30) Working 40% (12/30) High school ed or less 17% (5/30) LGBT 20% (6/30) Fagerstrom test score 5.4 (SD 2.1) Smokes more than half pkt cigs per day 53% (16/30) Smoked for 10+ years 93% (28/30) Use ecigs in past month 37% (11/30) 1 quit attempt in past 12 months 69% (20/29) Number of attempts to quit in 12 months 2.6 (SD 4.0)	Self-reported	Quit rates	Self-reported 30-day point prevalence quit rate for those who completed the 2-month follow-up was 20% (5/25) for Quit2Heal versus 7% (2/29) for QuitGuide (odds ratio [OR] 5.16, 95% CI 0.71–37.29; *p* = 0.10). The 30-day adjusted point prevalence quit rate was 17% (5/29) for Quit2Heal versus 7% (2/30) for QuitGuide (OR 3.87, 95% CI 0.57–26.16; *p* = 0.17). Quit 2 heal participants had improved outcomes for ‘internal shame’ cancer stigma, depression, and anxiety—not statistically significant.
Crawford et al., 2019 [26]	Patients	N = 569 participants eligible after phone screen, 283 (49.7%) attended the intake session and 207 enrolled/. 50% female, 69.6% white, 50.2% married, 64.7% educated below college grad., 48.3% employed, 12.6% stage 1 cancer, 7.2% stage 2, 21.7% stage 3–4, 21.3% remission, 37.2% not specified Mean age 58.48 (SD 9.44), Mean cigs/day 14.96 SD 8.23, FTCD 4.5 SD 2.13, years smoked 40.43 yrs SD 11.32.	Carbon monoxide breath samples monitored in-person at Weeks 0, 4, and 12.	Quit rates Side effects of medication Carroll 2019 [27] Quit attempts and smoking history. Past 24 h cigarette use, Positive and Negative Affect Schedule (PANAS). Data collection baseline, quit day, 4 weeks and 12 weeks. Fagerstrom test. Scholl 2019 [48] Quit rates 24 weeks varenicline.	Week 12, 73 participants (35.3%) had quit smoking and 107 participants (51.7%) reported taking >80% of the medication. More than 50% of patient were non adherent to treatment. Mood management in early treatment weeks may mitigate the effect of depression symptoms. A longer pre cessation period with varenicline may be needed to mitigate the effect of nicotine reward on adherence. Price 2017 No reports of depressed mood, suicidal thoughts, or CV events. Abstinence of smoking associated with improved cognitive function. Miele 2018 75.81 (SD 49.19) cigs smoked 7 days prior to quit visit. Week 12 n = 73 (35.3%) quit smoking. and n = 107 (51.7%) reported >80% adherence with meds. 62 of adherent (58%) quit smoking V 11 of those who were not adherent (11%) > x2 (1) 53.8, *p* < 0.001. With a Cut off <5 ppm quit 53% adherent V 14% non-adherent x2 (2) =32.6, *p* = 0.001. Logistic regression analysis (Predicting adherence) from baseline to Week 4: Age predictor OR 1.11, 95% CI 1.01–1.22 *p* = 0.030 (older smokers more adherent). Depressed mood was a predictor of adherence OR 0.38, 95% CI 0.17–0.84, *p* = 0.016. Side effect of vomiting and sleep problems greater in non-adherent participants. Decreases in the satisfaction from smoking (OR = 0.49, 95% CI = 0.26 to 0.93, *p* = 0.03) and increases in the toxic effects of smoking (OR = 4.73, 95% CI = 1.44 to 15.56, *p* = 0.01) predicted greater varenicline adherence. At week 12 (multivariate models re-run using 12-week pill count as a measure of adherence): depression approached significance. positive effect was a significant predictor; smoking-related variables measured by Cigarette Evaluation Scales (CES were no longer significant; increased vomiting was significant) Carroll 2019 [27] N = 119 attended all 4 person sessions. 81% adherent (n = 96). No significant predictive associations between PA and smoking (among either adherent or non-adherent participants). Among participants who adhered to varenicline, higher levels of smoking predicted higher NA which reducing or quitting smoking predicted lower levels of NA at next visit. Schnoll 2019 [30] Primary outcomes were 7-day biochemically confirmed abstinence at weeks 24 and 52. Point prevalence and continuous abstinence quit rates at weeks 24 and 52 were not significantly different across treatment arms (P’s > 0.05). Adherence (43% of sample) significantly interacted with treatment arm for week 24-point prevalence (odds ratio [OR] = 2.31; 95% confidence interval [CI], 1.15–4.63; *p* = 0.02) and continuous (OR = 5.82; 95% CI, 2.66–12.71; *p* < 0.001) abstinence. No differences between treatment arms on side effects, adverse and serious adverse events, and rates of high blood pressure (P’s > 0.05).
Foshee et al., 2017 [35]	Patients	Intervention group n = 48, vs. other group n = 44. Demographic data on those who completed follow up surveys Intervention N = 27 V other N = 25. Male n = 13 V n = 10, Diagnosis Malignant n = 13 V n =12, smoking history 20+ years n = 26 V n = 20, ready to quit N = 27 V m = 25 (100%)	Not reported	Quit rates	Those who received book less likely to quit. 26% of (n = 27) quit V 32% of n = 25 *p* = 0.63. Those who received book more likely to read it 77.8% V 52% *p* = 0.0563. Reading the book not associated with quitting: 29.4% of participants who read the book quit smoking by the end of the study compared with 33.3% who did not read the book (*p* = 0.81).
Ghosh et al., 2016 [37]	Patients (Veterans)	Patient demographics—total number not clear (total eligible n = 114). N = 24 consented, N = 14 attended cessation classes Intervention group N = 6 V control N = 8. Age (M) Intervention 59 years V 61 years control. Gender Male, Ethnicity Black African American, Education 1–3 years at college (both groups) Quit attempts previously = 0 control vs. 2 incentive group. Saving/yr if quit $1200 Intervention V $1000 control.		Quit rates	N = 2 quit at 6 months intervention group. (n = 2 quit at 3 months control group but lost to follow up). SF QOL Intervention group Scores 4 weeks 34, 3 months 32, 6 months 35.5 (max 48). Control group 30 days 32.6, 3 months 30. Quit rates at 1 month were not sustained at 3 months in the control group. Veterans’ mobile population, travel and distance for follow up could have led to higher rates of non-enrollment/lost to follow up.
Krebs et al., 2019 [38]	Patients Standard care n = 18, Quit IT n = 20	N = 38 patients randomised into pilot study. N = 15 (40%) aged 50 to 59 years. n = 27 (71.1%) female. n = 35 (92.1%) white, n = 36 (94.7%) non-, n = 12 (31.6%) lung cancer. n = 17 (45%) college education, n = 24 (63%) (used tablet, smoking since diagnosis n = 30 (79%) decreased) quit attempts in past year yes once n = 9 (24%), yes, more than once n = 20 (53%) Fagerstrom score n = 38 3.68 (SD 2.2), years smoking 36.7 yrs (SD 12.6), baseline cigs/day 12.34 (S 14.7).	Biochemical verification with salivary cotinine assays Patients using NRT/ecigs—breath sample taken in person to test for expired CO.	Quit rates Feasibility of use	At 1 month n =24 completed (Standard Care n = 11, Quit IT n = 13) of which n = 18 female (67%) n = 19 (63%) decreased smoking. Quit attempts No = 4/24, yes once 7/24, yes more than once 13/24. Confirmed abstinence was higher in the QuitIT arm, with 30% (4/13) of the sample reporting abstinence versus 18% (2/11) in the SC arm.
Park et al., 2020 [43]	Patients	Intensive Treatment N = 153 V standard care n = 150. Treatment age (median IQR) 59 (52–65) V 57 (52–65). Treatment gender Make 43.1% (66/153) UC 44.7% (67/150). Treatment ethnicity/race White 83% (127/153) V UC White 85.3% 128/150. Treatment Married 58.2% 89/153, V UC 51% 75/150, employed Treatment 38.3% (57/153) V UC 49.7% (73/150), Home smoking rule no smoking anywhere 53.3% 80/150 V UC 44.5% 65/146, smoking in some places 27.3% V UC 24.7%, smoke anywhere 19.3% V UC 30.8%.	7-day tobacco absence biochemically confirmed at 3 and 6 months. Self-report between 3 and 6 months.	Smoking rates Quit rate	Treatment V UC 6 months n = 51 V n = 29 OR 1.92 95% CI 1.13–3.27, *p* = 0.02. 3 months n = 46 V n = 28 OR 1.72, 95% CI 1.00–2.96 *p* = 0.048. Sustained absence at 6 months n = 35 V n = 17 OR 2.15 95% CI 1.14–4.05 *p* = 0.02. The median number of counseling sessions completed was 8 (IQR, 4–11) intensive treatment group. 97 intensive treatment participants (77.0%) vs. 68 standard treatment participants (59.1%) reported cessation medication use (difference, 17.9% [95% CI, 6.3–29.5%]; odds ratio, 2.31 [95% CI, 1.32–4.04]; *p* = 0.003).
Rettig et al., 2018 [45]	Patients	n = 19 Intervention V n = 10 UC. Age (both groups), years, median (IQR) = 55 (52–62). Gender Intervention Male = 11 (58%) V UC n = 7 (70%) Race Int white n = 13 (68%) V UC n = 5 (50%). married Int n = 8 (42%) V UC n = 3 (30%). Pack year smoking Int 45 (IQR 27–68) V UC 50, (IQR 30–50) > Cig smoked day Int Med 20 IQR 20–30 V UC Med =20, IQR 20–30. Use e cigs Int Yes n = 11 (58%) V UC n = 8 (80%).	CO monitor	Quit rates Cigs smoked	Participants in the intervention group were significantly more likely to abstain from cigarette smoking than those in the control group at week 8, (74% vs. 30%); *p* = 0.046; At 12 months, Participants in the intervention group smoked significantly fewer cigarettes per week at week 8 (median 0 vs. 10; *p* = 0 0.04), smoked fewer total cigarettes during weeks 1 to 8 (median 49 vs. 156; *p* = 0.09), and had a greater reduction in number of cigarettes smoked per week at week 8 compared with baseline (median 228 vs. 214; *p* = 0.28). Assignment to the intervention group was associated with nearly 5-fold higher odds of smoking abstinence (unadjusted OR 4.83; 95% CI 1.31–17.76).
Simmons et al., 2020 [46]	Patients	SRP n = 191 V UC n = 190. Females SRP n = 99 V UC n = 99, Age M 55.0 (11.0) V UC 55.2 (10.6), White SRP n = 181 (95.3%) V UC n = 171 (90.0%). No cigs/day SRP 21.6 (9.5) V 19.7 (9.1). No year smoking 35 (12.5) V 34.3 (11.7). Fagerstrom 5.3 (2.1) V 5.0 (2.2), confident abstinence at 6 months n = 91 (48.4%) V 106 (56.7%). Quit self-efficacy (9–45) SRP 38.0 (7.6) V UC 38.3 (7.6).	CO monitor	Quit rates	Quit at 2 months (if married/living together) 75% V 71% UC OR 2.23 (95% CI 1.01–4.90). At 6 months married/living together 78. % V 66.4% UC OR 1.86 (95% CI 0.97–3.56). At 12 months n = 272: abstinence rates SRP = 68% V 38% UC (*p* = 0.38) OR 1.24, (95% CI 0.77–2.00). No difference if married/partner (*p* = 0.84).
Smaily et al., 2021 [50]	Patients	N = 91 approached in participate. N = 62 eligible and n = 56 participated. Usual care n = 29 V intervention group n = 27. Age 61.9 (12.3) UC V 59.9 (9.8) SIG. M:F ratio 19:10 UC V 18:09 SIG. Tobacco use 2.1 packs/day (0.8) V 2.2 (0.9) SIG. Mean FTNS 6.4 (2.4) V 7.1 (2.6)	Self-report	Quit rates	Non-significant impact. Cessation rates 3 months 57.1 V 57.7% *p* = 0.96; 6 months 42.9% V 24% *p* = 0.148. 12 months 33.3% V 20.8% *p* = 0.318.
**Observational studies**					
**Study ID**	**Outcome Groups**	**Participant Demographic (Age, Sex, Smoking History If Included. Number Cigs Smoked; Prev. Quit Attempt)**	**Biochemical Verification**	**Outcome Measure**	**Outcomes**
Abdelmutti et al., 2019 [24]	Patients	Between April 2016 and March 2018, 13,617 new patients (62%) were screened for their smoking status. Of those patients, 1382 (10%) were identified as current smokers, and 532 (4%) reported having quit within the preceding 6 months.	Not reported in paper	Referral Rates	Referral rates. n = 380 (20%) accepted referral. n = 1534 (80%) refused referral. Of those who were referred, n = 131 (34%) internal referral and n = 248 (65%) external referral. Guiliani 2019 [37] N = 17842 patients attended. N = 5343 during 6-month pre-intervention, n = 7116 8 months implementation, n = 5383 6 months post-implementation. Only 36 screened using paper (1%). Referrals to smokers increase 18.6% 58/311 to 98.8% 421/426 *p* < 0.01. Accepted referrals decreased 41% 24/58 to 20.4% 86/421. Pre-post questionnaires 29.7% 83/279 returned pre and 41.9% 288/686 post-implementation. 29% in pre cohort still smoking (24/83) and 83/288 28.8% post-intervention. 37 of 88 (42%) in pre cohort vs. 101/288 (35.1%) post-implementation stopped smoking in past 4 weeks. Referrals increased from 19% to 99%. Screening increased from 44% to 66%.
Cinciripini et al., 2019 [33]	Patients, employees & patients with no cancer history	N = 3245 smokers, N = 2652 smokers & cancer,1588 (48.9%) were men, 322 (9.9%) were of black race/ethnicity, 172 (5.3%) were of Hispanic race/ethnicity, and 2498 (76.0%) were of white race/ethnicity. Mean (SD) age was 54 (11.4) years; Fagerström Test for Cigarette Dependence score, 4.41 (2.2); number of cigarettes smoked per day, 17.1 (10.7); years smoked, 33 (13.2); and 1393 patients (42.9%) had at least 1 psychiatric comorbidity.	Expired carbon monoxide levels monitored at all in-person visits.	Quit rates	Smoking status was assessed at 3, 6, and 9 months. 7-day point prevalence abstinence at 9 months, defined as self-report of no smoking (not even a puff) during the previous 7 days. Abstinence (overall) 45.1% at 3 months, 45.8% at 6 months, 43.7% at 9 months. Abstinence no different if cancer/no cancer. 3 months RR 1.03 95% CI 0.93–1.16 *p* = 0.55; 6 months RR 1.05 95% CI 0.94–1.18 *p* = 0.38, 9 months RR 1.10 95% CI 0.97–1.26 *p* = 0.14. Head and neck cancers abstinence rates higher at 9 months RR 1.31 95% CI 1.11–1.56. *p* = 0.001.
Conlon et al., 2020 [47]	Patients: N = 493. smokers n = 183, ex-smokers n = 310	N = 1245 patients with head-and-neck cancer attended the Dental Oncology clinic. N = 493 ever smoked enrolled in the study. N = 183 smoking at enrollment, N = 310 ex-smokers who had quit. 493 ever-smokers enrolled in the study (96.1%). Age at enrolment was 66 years (37–96), 76.9% males, age smoking median 16 years (IQR 4 to 60).	Self-reported	Quit rates, quit attempts	85.8% interested in quitting and 70.5% considering quitting within next 30 days. Current smokers (n = 35, 19.1%) reported quit attempt less than 1 month or up to 1 month; 14.8% (n = 27) had been able to quit for 3–6 months and 7–12 months (each). In prior quit attempts, many current smokers had used no cessation aids, choosing to quit “cold turkey” (48.5%), although 23.8% had also tried nicotine replacement therapy.
Davidson et al., 2018 [48]	Patients	N = 13240 new patients and n = 10341 (78%) screened for tobacco use. 18% (n = 1866) current/recent smokers. Of 1866, n = 1507 (81% of 1866) advised of cessation benefits, n = 1499 (80%) offered referral to smoking cessation. n = 211(11%) accepted referral, n = 51 (3%) smokers enrolled in programme.	Not reported	Referrals	Increase in referral rates 77% offered referral of which 9% were smokers and 2% enrolled.
Day et al., 2020 [34]	Patients	n = 117. Male N = 80 (68.4%), Age median 57 years, IQR 50–61. n = 105 white (89.7%), N = 67 married (57.3%), education advanced degree/degree (n = 58 49.5%), yrs smoked median 39, IQR 25–52 Med cigs/day 20 IQR 20–30.	Self-reported smoking status and carbon monoxide testing.	Quit rates	Abstinent N = 49 (42%) V non abstinent N = 68 (58%). Male abstinent 63.3% (n = 31) Vs Female abstinent 72.1% N = 49. Abstinent at 9 months N = 49. There were no significant differences in sociodemographic or tumor characteristics of patients according to abstinence at 9 months 90% congruence self-report and CO testing.
Phillips et al., 2020 [43]	Patients	N = 82 smokers identified. Male 52.4%. Caucasian 98.8%. Age 62.4 yr (SD 7.2). Pack years 51.9 (SD 25.7). 73.2% CO confirmed smoking at time of surgery. Quit by surgery (n = 60) V smoking at surgery (n = 22). N = 60 quit by surgery, N = 22 smoking at time of surgery. 78.3% of quit by surgery met with TTS (n = 47/60) V 72.7% (14/22) smoking at surgery.	CO breath levels pre-op and day of surgery	Quit rates	N = 63 met with TTS. N = 60 quit at time of surgery. N = 22 smoking at time of surgery. Smoking cessation 6 months = 55.3%. 70.4% in quit at time of surgery V 18.1% smoking at surgery (*p* < 0.0001). 1 year 55.6% not smoking. 64.4% quit at surgery V 33.2% smoke at surgery (*p* < 0.025). 2 years smoking cessation 51.7% Quit at surgery 55% V smoke at surgery 44% (*p* = 0.63). Deaths and loss to follow up at each time point. Quit attempts—intermittent stop and start over 2 years. Those who quit before surgery and never smoked again V unable to quit before surgery 62% V 18% (*p* < 0.001).
Ramsey et al., 2020 [44]	Patients	Sample demographics from records (n = 474,674), Female 282,283, Male 192,197. Age range 18–118. Total Smokers Rural = 9751 Urban = 52,369.	Not clear	Smoking prevalence	Smoking prevalence significantly higher in rural clinics (20.7%) compared to urban clinics (13.9%). a lower proportion of smokers received smoking treatment in rural clinics (9.6%) than in urban clinics (25.8%). Patients were more likely to receive cessation treatment in cancer clinics that had implemented versus clinics that had not yet implemented the smoking cessation module = 31.2% V 17.5%
**Mixed methods and qualitative studies**					
**Study ID**	**Outcome Groups**	**Participant Demographic (Age, Sex, Smoking History If Included. Number Cigs Smoked; Prev. Quit Attempt)**	**Biochemical Verification**	**Outcome Measure**	**Outcomes**
Abdelrahim et al., 2018 (Qualitative) [49]	Patients	8 Male; 3 Female, 7 smokers, 4 Ex-smokers (3 F, 1 M). Median age: 56 yrs, (44–70). n = 4 quit at diagnosis or prior to. Smoking status 7 Smokers: 4 Ex—Smokers (3 F, 1 M) Age median 56 years (44–70 years) All participants were between 12 months–4 years from completion of treatment and in the surveillance period.	Not reported in paper	(1) the individual’s relationship with smoking before and after a diagnosis of head and neck cancer, (2) attempts at quitting both successfully and/or unsuccessfully and what influenced these, (3) healthcare provision and knowledge (4) experience of smoking cessation support services.	Themes described guilty habit of smoking, perceived barriers to quit, teachable moment of a diagnosis and social motivation to both smoke and quit. ‘Guilty habit’ represented the knowledge that smoking was ‘wrong’ and socially stigmatised, difficulty in adjusting to a life without tobacco and cigarettes. Feelings of guilt and self-blame when smoking after treatment for cancer. Smoking relapse was common. It is unlikely that a simple information giving exercise would impact quitting. Sustained support and encouragement needed. Barriers to quit refusal did not indicate refusal, suggest asking again, wanted ownership of ability to quit. Cost was as barrier, stress was a barrier, cancer treatments cited as a barrier—slowness and challenges eating often relieved by a cigarette. Social motivation—Boredom noted for those with less social supports. However, social support was complex and could encourage smoking. Sometimes exacerbated by the social isolation and unemployment that is often a feature of living with HNC. Teachable moment identified by participants who quit—and in many who hadn’t but wish to do so. Shock of diagnosis a motivator as pain in mouth from cancer identified. Understanding the link between smoking head and neck cancers can be paramount. Cessation provisions and effects: For those who had used smoking cessation supports, an abrupt cessation of the smoking cessation supports gave a feeling of being ‘left to their own willpower to quit; some patients experienced unwanted side-effects of pharmacotherapy.
Charlot et al., 2019 [32]	Patients	N = 18 Age (50 to 70 years) Mean 64.2 yrs, SD 8.0, Females 61.1% (n = 11), non-white 44.4% (n = 8), English lang (94.4%) n = 17, education high school or less 44.4% n = 8, not working 78.8% n = 14, income <$10,000 27.8% (n = 5), active cancer treatment Y 44.4% n = 8, cig smoked/day 10 or less 55.6% n = 10, 11–20 44.4% n = 8.	None reported	Quit rates, focus group feedback feasibility	Reduction in cigarettes smoked. Cigs smoked baseline 75.1 mean weekly intake. Week 8 = 50.1. 3 month follow up reduced to 44.3. N = 12 completed final data collection. Positive outcomes with Mindfulness smoking cessation programme. supportive, satisfied with mindfulness ‘ our bible’. Winter weather, late time of day (4–6 pm) made attendance less positive due to traffic, lack of parking, concerns walking in the dark.
McDonnell et al., 2016 [40]	Patients and family members	N = 8 dyads (8 patients and family members). 100% Caucasian. Patients 100% male. Partners 100% female. n = 5 patients known neoplasm at pre-op baseline. Median age patients 58 years. 6 of 8 dyads married and lived in homes with no indoor smoking restrictions. Median age family 49 years. 6 of 8 dyads married and lived in homes with no indoor smoking restrictions.	Self-report	Feasibility Quit rates	Low recruitment—50 patients screened. N = 16 reported one quit attempt. n = 5 quit attempt in past year. All reported one quit attempt. n = 5 quit attempt in past year. No attempt to stop a patient smoking, rated smoking as very important to them. Stopping smoking important but low confidence at outset. Face to face meetings associated with adherence. Two additional booster meetings with financial incentive to boost adherence. Challenges included time as recruitment over the summer. Private space to meet. Due to changes in university, lack of referrals due to team members considering it was too late to refer to smoking cessation services—fidelity was retained to reduce team bias through a brief orientation to all new members of the team. Recruitment 44% lower than anticipated despite team support, financial incentives, assurance of privacy,—limitation requirement of family member to participate. Exit interviews identified pre-op timing as ideal, telephone interactions difficult due to competing life issues. No attempt to stop a patient smoking, rated smoking as very important to them. Family members are less confident than partners in own ability to quit.
Smith [51] et al., 2019	Patients	N = 77 eligible and n = 64 consented (83%). Current smokers n = 29. Age 60.1 (SD 7.6) V former/never smoked 60.2 (8.9). Male n = 24 V n = 30. Smoking pack years M SD 45 (24.7) V 26 (29.5).	CO readings	Quit rates	Quit rates: 14/26 (53.8) 3 months 11/26(42.3%). 7-day point prevalence 1 month 18.25 (72%) 3 month follow up 16/24 (66.7), cessation during radiotherapy 19/24, (79.2%). N = 6 reported as smokers. 5 themes. Teachable moment in cancer diagnosis, use of personal willpower and cessation aids, psychosocial environment, the relationship with alcohol and drugs, and the interaction between health knowledge and beliefs of cancer and smoking. Patients who consumed alcohol less confident quitting and lower stage of change.
**Quality Improvement Studies**					
**Study ID**	**Outcome Groups**	**Participant Demographic (Age, Sex, Smoking History If Included. Number Cigs Smoked; Prev. Quit Attempt)**	**Biochemical Verification**	**Outcome Measure**	**Outcomes**
Gali et al., 2020 [36]	Patients and families	n = 368 smokers identified at screening (6%) of the 99% patients screened.	Self-report smoking status	Referral rates Quit rates	n = 44 6 months follow up. n = 9 quit/tobacco-free (20% of those who engaged). By 11 months service expanded to 11 clinics. Average number of smokers referred/per week increased from 3 to 35. n = 600 referred. n = 181 (30%) engaged in treatment n = 273 74% contacted via phone follow up, of which n = 90 (33%) engaged with the service (33%). n = 42 (47%) selected pharmacotherapy or behavioural change. n = 48 (53%) selected combination of treatments. n = 9 10% requested quitline information. N = 61 (68%) involved family/individual/group counselling services. n = 15 completed 3 sessions (25%). N = 59 (66%) NRT, n = 48 refills provided. n = 24 (27%) consultation for varenicline or bupropion.
Nolan et al., 2019 [41]	Patients and employees	N = 45 patients self-reported smoking—data collected, and N = 10 interviews with patients.	Not reported	Quit rates/attempts Feasibility	n = 45 patients. n = 15 smoking at intake call reported quit smoking by time of breast clinic appointment. n = 30 smokers at time of visit 23 (76%) referred to Nicotine Dependency Clinic (NDC). Significant increase to pre-intervention (29% *p* < 0.0001). Of those referred, 17 (74%) attended NDC consult—an increase from pre-intervention (41% *p* = 0.026). Qualitative data from n =10 patients. 5 referred to NDC. 4 of 5 planned to attend. System-level interventions—patients recalled stop smoking conversations related to breast reconstruction. Intervention factors identified by patients—gratitude that smoking not discussed front and centre as their primary concern was diagnosis. No patient surprised that smoking was discussed or offered NDC consult. Many could not describe the NDC consult.
**(b)**					
**Study ID**	**Outcome Group**	**Participant Demographic**	**Biochemical Verification**	**Outcome Measure**	**Outcomes**
Ma et al., 2016 [39]	Employees	N = 117 patient chart reviews. N = 15 care providers surveyed.	Self-reported in charts	Referral quit services and knowledge	13% of staff were aware of tobacco control services. 28% documented. Pre-intervention N = 54 charts identified 6 to 13 smokers/month. Median 8. Pre-intervention lack of knowledge of services and unawareness of smoking cessation services in the community. N = 2 of 15 staff were aware of any resource in the community. Post-intervention 100% of providers could name one cessation resource and 88% felt the intervention prompted discussions.
Nolan et al., 2019 [41]	Employees	N = 12 qualitative interviews with providers	Not reported	Factors associated with service plan	Provider interviews N = 12 identified systems-level factors—important that smoking stopped before reconstructive surgery (4 to 6 weeks stopped). This surgery used to facilitate discussions about smoking. 2nd theme was perception of team roles—an assumption that previous providers had already discussed smoking status with a patient. Personal decision factors: factors identified as barriers to discussion quitting included: patients who were seen as light smokers (a few a day), advanced cancer, perception of emotional instability, presence of comorbid conditions, geographical distance. Facilitators included: patients’ interest in reconstructive surgery, no script for this, but all providers discussed own experiences with family, friends, experiences from other patients who had identified tobacco cessation counselling positive. Intervention level factors from providers included: a lack of knowledge among providers about what happens at NDC consults (same as patients).

**Table 4 ijerph-19-17010-t004:** Summary of inventions processes and outcomes.

Study	Setting	Enroll	SC Local	SC National	EHR	Multi Lang	Smoke Status Record All Visits	Opt Out	Staff Training	SC by Expert Oncology HCPs	Follow Up Face to Face	Follow Up Phone/Email	NRT	Pharmaco–Therapy	Family Option	Success	Intervention ≥ 6 Months Duration
Abdelmutti et al., 2019 [24] Giuliani et al., 2019 [25]	Hospital/cancer clinics	Newly diagnosed in patients	Y	Y Ottawa model	Y	Y		Y	Y		Y	Y	Y	Y		Y referral	Y
Abdelrahim et al., 2018 [49]	Hospital clinics	Patients who had completed treatment	Y	Y NHS Stop smoking service												NA	Y
Bricker et al., 2020 [31]	Social media recruit from two cancer centres	Diagnosed within past 12 months or currently or planning to receive treatment	Y	Y QuitGuide app			Ongoing app engagement									NS	Y
Charlot et al., 2019 [32]	Hospital medical centre oncology clinics	Had a cancer diagnosed. Not currently receiving treatment and more than 6 months diagnosed.	Y							Y	Y 8 week x 2 h group					NS	N
Cinciripini et al., 2019 [33]	Cancer centres	All referred to Cancer centre. (can include those without cancer)	Y								Initial 60–90 min. 6–8 follow up 30–45 minus over 8–12 weeks—95/% via phone	Y	Y	Y	Y	Y	Y
Conlon et al., 2020 [47]	Dental oncology clinic at National Cancer centre	Newly diagnosed patients prior to commencing treatment						Y		Y	Follow up every 1–2 weeks during active radiation therapy. And 4, 8, 12 weeks, 6 months and 1 year post treatment	?	Y	Y		NS	Y
Crawford et al., 2019 [26]/Carroll et al., 2019 [27]/Miele et al., 2018 [28]/Price et al., 2017 [29]/Schnoll et al., 2018 [30]	Hospital	Within 5 years diagnosis or cancer treatment	Y							Y. Assessment 60 min at commencement	Y week 0, 1, 4, 8, 12, (In person 0, 4, 12) (20 min)	Y (week 8)	Y	Y		Y	Y
Davidson et al., 2018 [48]	Hospital	All patients register at first consultation	Y	Y Ottawa model	Y		Y			Y within 2 weeks. follow up. Counselling and referral to external service.	Y as needed—have pager	Y—automated phone or email for 6 months. Additional follow up by nurse as needed.	Y	Y		Y referrals	Y
Day et al., 2020 [34]	Hospital	All patients with histopathological confirmation of cancer and received primary treatment	Y TTP								Y Initial 60–90 min.	Y—6 to 8 follow up 30–45 minus over 8–12 weeks.—95% via phone	Y	Y		?	Y
Foshee et al., 2017 [35]	Hospital	Patients attending Dept of Otolaryngology Medical charts checked for cancerous or non cancerous tumors	Y									Y (2 weeks up to 6 months. And 6 months to 1 year).				N	Y
Gali et al., 2020 [35]	Hospital	All patients attending	Y	Y Quitline number	Y	Y	Y	Y	Y	Y	Y—2 week trial (free)	Y—Individual/group/virtual/phone—all offered	Y	Y—expediated service for prescribing and dispensing in hospital or local dispensary		Y referrals	Y
Ghosh et al., 2016 [37]	Hospital	Attending clinic for evaluation or treatment of malignant/pre malignancy lesion. All diagnosed cancer	N	Y							Y Attend 3 classroom sessions.	Y by phone after 30 days enrolled and 3 months, 6 months				NS	Y
Krebs et al., 2019 [38]	Hospital	Diagnosed within 6 months, scheduled for surgery. Consent following surgery as inpatient.	Y Quit-IT						Y	Y	Y—up to 4 counselling sessions—either bedside or phone. IPAD App game for 1 month post hospitalization	Y patient choses either in person or phone sessions				Y quit	N
Ma et al., 2016 [39]	Hospital	Newly diagnosed patients at clinic	Y	Y Community resources/quit lines			Y	Y	Y	Y	Y—written information provided					?	N
McDonnell et al., 2016 [40]	Hospital	Patients awaiting surgery	Y						Y	Y	Y four face to face visits pre op and post op called ‘boosters’.	Y—and home visits made to 2 family members			Y	NS	Y
Nolan et al., 2019 [41]	Hospital clinic	Patients 1st visit with diagnosis or start treatment	Y				Y	Y		N	Y Initial 45 min consultation.	Y up to 3 phone call follow ups	Y			Y referrals	Y
Park et al., 2020 [42]	Hospital	Patients undergoing cancer treatments			Y					Y	Y—Option if patients selected to attend hospital	Y—4 weekly telephone follow up to reduce burden. Patient could opt for in person. 3 booster sessions offered	Y	Y—12 week no cost		Y quit	Y
Phillips et al., 2020 [43]	Hospital	Pre operative surgery. Patients reviewed initial surgery consultation	Y				Y				Y—As needed counselling. 1 h sessions (evaluations up to 24 months)	Y—as option and as needed by patient	Y	Y		Y	Y
Ramsey et al., 2020 [44]	Oncology clinics	Patients attending	Y	Y link to helplines and resources	Y		Y	Y	Y		Y		Y	Y		Y referrals	Y
Rettig et al., 2018 [45]	Hospital	Patient planned radiotherapy for 5 or more weeks	Y	Y enrolled to NCI app				Y		Y	Y 8 weekly visits	Y text messaging	Y	Y		Y	Y
Simmons et al., 2020 [46]	Hospital	Patients recently diagnosed and initiating treatment	Y								Y	Y—mailed 7 booklets	Y	Y		Y	Y
Smaily et al., 2021 [50]	Hospital	Patients admitted for biopsy, start treatment, or surgical management	N	Y					Y		N	Y—after 5 weeks enrolment.	Y	Y		NS	N
Smith et al., 2019 [51]	Hospital	Patients newly diagnosed and commenced treatment	Y Surveys and interviews								.					Y	N

SC local = smoking cessation local programme; SC National = smoking cessation national programme; Multilang = available in multiple languages; Opt out = programme developed and all patients included. Anyone who did not want to participate had to opt out; Staff training = availability of training on smoking cessation for staff for new intervention; SC by expert oncology HCPs =smoking cessation provided by oncology healthcare professionals in oncology dept; NRT = Nicotine replacement therapy; Family option = Families specifically included in intervention; Success = was there a positive outcome from intervention; Y = yes, N = no.

## Data Availability

Data extraction tables can be provided if requested.

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
