# Peer review of "Systematic Review of Smoking Cessation Interventions for Smokers Diagnosed with Cancer"

_ijerph, 2022, doi:10.3390/ijerph192417010_

Round 1
Reviewer 1 Report
I would like to highlight a few points that the authors should correct.
1) For example, Abdelmutti should be Abdelmutti et al. (2019). This style should be corrected through the article.
2) Moonshot should be opened.
3) The objective should be formulated differently ("A review of the international evidence base was undertaken as part of a funded study to develop a smoking cessation pathway tailored for people with lung, cervical, head and neck and breast cancers").
What did you really want to explore here?
4) The review question should be at the end of Introduction, why is it in the Material and Methods paragraph?
5) Figure 1: Why was N=4126 removed? Please, specify.
6) Why have studies with a duration of less than 6 months also been included in the review? Those studies should be eliminated.
Reviewer 2 Report
In review of your manuscript, "A Rapid Systematic Review of Smoking Cessation Inventions for Smokers Diagnosed with Cancer", I found it well written and a great potential enjoyment for the readers. My two requests: A grammar review of the manuscript and trying to understand "rapid" systematic review. I feel it is rather thorough. Either flush out more thoroughly what "rapid" means in this context, or consider removing.
Round 2
Reviewer 1 Report
Thank you, the article has improved. I would ask you to take into account how long the intervention has been monitored and to separate into a separate group those who had been monitored for less than 6 months and to mention this matter in limitatios.
Furthermore, what is this referring to: "A review 82 of the international evidence base was undertaken as part of a funded study to develop a 83 smoking cessation pathway tailored for people with lung, cervical, head and neck and 84 breast cancers. "
Author Response
Please see the attachment and the revised document will be uploaded into the system.
Many thanks
